# Characterization of Two Aldehyde Oxidases from the Greater Wax Moth, *Galleria mellonella* Linnaeus. (Lepidoptera: Pyralidae) with Potential Role as Odorant-Degrading Enzymes

**DOI:** 10.3390/insects13121143

**Published:** 2022-12-12

**Authors:** Ricardo Godoy, Ignacio Arias, Herbert Venthur, Andrés Quiroz, Ana Mutis

**Affiliations:** 1Programa de Doctorado en Ciencias de Recursos Naturales, Universidad de La Frontera, Temuco 4811230, Chile; 2Departamento de Ciencias Químicas y Recursos Naturales, Facultad de Ingeniería y Ciencias, Universidad de La Frontera, Temuco 4811230, Chile; 3Carrera Bioquímica, Universidad de La Frontera, Temuco 4811230, Chile; 4Centro de Investigación Biotecnológica Aplicada al Medio Ambiente, CIBAMA, Universidad de La Frontera, Temuco 4811230, Chile

**Keywords:** beekeeping, bioinformatics, olfaction, sexual communication

## Abstract

**Simple Summary:**

The greater wax moth, *Galleria mellonella* Linnaeus (Lepidoptera: Pyralidae), is a ubiquitous pest of the apicultural industry. We identified two novel aldehyde oxidase genes through transcriptomic analysis (*GmelAOX2* and *GmelAOX3*) that are related to its olfactory system. *GmelAOX2* is part of the clade with odorant-degrading enzyme function and shows sex-biased expression, and both *GmelAOX2* and *GmelAOX3* are more highly expressed in male antennae rather than female antennae. These enzymes have a crucial role in metabolizing sex pheromone compounds as well as plant-derived aldehydes, which are related to honeycombs and the life cycle of *G. mellonella*.

**Abstract:**

Odorant-degrading enzymes (ODEs) are proposed to degrade/inactivate volatile organic compounds (VOCs) on a millisecond timescale. Thus, ODEs play an important role in the insect olfactory system as a reset mechanism. The inhibition of these enzymes could incapacitate the olfactory system and, consequently, disrupt chemical communication, promoting and complementing the integrated pest management strategies. Here, we report two novel aldehyde oxidases, AOX-encoding genes *GmelAOX2* and *GmelAOX3*, though transcriptomic analysis in the greater wax moth, *Galleria mellonella. GmelAOX2* was clustered in a clade with ODE function, according to phylogenetic analysis. Likewise, to unravel the profile of volatiles that *G. mellonella* might face besides the sex pheromone blend, VOCs were trapped from honeycombs and the identification was made by gas chromatography–mass spectrometry. Semi-quantitative RT-PCR showed that *GmelAXO2* has a sex-biased expression, and qRT-PCR indicated that both *GmelAOX2* and *GmelAOX3* have a higher relative expression in male antennae rather than female antennae. A functional assay revealed that antennal extracts had the strongest enzymatic activity against undecanal (4-fold) compared to benzaldehyde (control). Our data suggest that these enzymes have a crucial role in metabolizing sex pheromone compounds as well as plant-derived aldehydes, which are related to honeycombs and the life cycle of *G. mellonella.*

## 1. Introduction

In insects, mating or threat avoidance, as well as host and food seeking, are mainly results of the interaction between volatile organic compounds (VOCs) and a well-tuned olfactory system [1,2]. The transport, transduction, and degradation of VOCs are carried out by olfactory proteins, such as odorant-binding proteins (OBPs), odorant receptors (ORs), and odorant-degrading enzymes (ODEs) [3,4,5,6]. Briefly, chemical cues enter through cuticular pores placed in hair-like structures called sensilla on antennae, towards the sensillar lymph where OBPs bind and transport molecules to the ORs. Once receptors are activated, a transduction signal is started, unleashing insect behavioral response. Finally, ODEs rapidly degrade the odorant stimuli in order to avoid the accumulation of compounds in the peripheral space, leading to rapid signal termination [4]. This allows the recovery and maintenance of the sensitivity of the olfactory system by its resetting on a millisecond (ms) timescale, and thus receive new chemical signals [7]. Noteworthy, there are different families of ODEs involved in stimuli deactivation, such as carboxylesterases (CXEs), aldehyde oxidases (AOXs), glutathione-S-transferases (GSTs), and cytochrome P450 (CYPs). Particularly, CXEs and AOXs have been studied as ODEs [8]; in fact, the first ODE (a CXE) was classified as a pheromone-degrading enzyme (PDE), being specifically present in male antennae of moth *Antherea polyphemus* Cramer [9]. This PDE was able to rapidly degrade (in an estimated half-life of 15 ms) the sex pheromone (*E*,*Z*)-6,11-hexadecadienyl acetate according to in vivo and in vitro assays [10]. Another PDE was identified in the scarab beetle *Popillia japonica* Newman, being only expressed in male antennae. The authors reported that both the native and recombinant enzymes showed preference against the sex pheromone, (*R*)-japonilure, rather than its enantiomer, (*S*)-japonilure (behavioral antagonist) [11]. On the other hand, the first AOX in moths was characterized in the tobacco hornworm *Manduca sexta* Linnaeus, and reported as a dimer from column chromatography with an estimated molecular weight of 295 kDa [12]. This AOX was classified as a PDE (MsexAOX) because it had nearly 60% greater expression in male than female antennae. Additionally, kinetic parameters of *MsexAOX* showed a preference for the pheromone compound, (*E*,*Z*)-10,12-hexadecadienal (bombykal) (K_m_ 5.4 µM), compared with other VOCs, such as propanal (K_m_ 6.8 µM) and benzaldehyde (K_m_ 225.1 µM) [12]. Complementary, Merlin et al. [13] proposed that the *MbraAOX*-encoding gene from cabbage armyworm *Mamestra brassicae* Linnaeus is active at the sensillar lymph level and that its expression is restricted to olfactory sensilla (i.e., Sensilla trichoidea, Str I) through in situ hybridization (ISH). In functional terms, these enzymes can transform aldehyde-type semiochemicals into inactive forms, such as carboxylic acids [14]. Nowadays, several moth species use a blend of chemicals as sex pheromones, where aldehydes can act as either major or minor components. For example, (*Z*,*Z*)-7,11-hexadecadienal, (*E*)-11,13-tetradecadienal, and (*E*,*E*,*Z*,*Z*)-4,6,11,13-hexadecatetraenal act as major pheromone components for the moths citrus leaf-miner *Phyllocnistis citrella* Stainton, the eastern black-headed budworm *Acleris variana* Fernald, and the promethea moth *Callosamia promethea* Drury, respectively [15,16,17]. On the other hand, aldehyde-based pheromones can act as minor components, such as (*E*)-10-hexadecenal, (*E*,*E*)-8,10-dodecadienal, and (*E*,*Z*)-6,11-hexadecadienal for the legume podborer *Maruca vitrata* Fabricius, the codling moth *Cydia pomonella* Linnaeus, and *A. polyphemus*, respectively [18,19,20]. Noteworthy, aldehydes can also act as behavioral antagonists. For instance, females of *Bombyx mori* Linnaeus emit bombykal in their pheromone blend in addition to bombykol as a behavioral antagonist [21].

The fact that *A. polyphemus* and *B. mori* use aldehydes in their life cycle supports the presence of AOXs that could metabolize these semiochemicals. Thus, Rybczynski et al. [22] identified antenna-specific aldehyde oxidases (AOXs) in antennal extracts from *A. polyphemus* and *B. mori*, which were more abundant in males than females through polyacrylamide gel electrophoresis (PAGE). Likewise, Zhang et al. [23] found four putative AOX genes in the rice leaf-folder *Cnaphalocrocis medinalis* Guenné through transcriptomic analysis. The authors reported that *CmedAOX2*-encoding transcript was more expressed in male than female antennae, indicating a putative degradation role for the sex pheromone blend, (*Z*)-11-octadecenal and (*Z*)-13-octadecenal. On the contrary, there are some exceptions where AOXs can degrade other aldehyde-based pheromone compounds, with no involvement of sex-biased expression. For instance, recent studies in the cotton bollworm *Helicoverpa armigera* Hübner identified six full-length AOX genes, from which *HarmAOX2* was suggested as PDE for inactivating (*Z*)-11-hexadecenal and (*Z*)-9-hexadecenal through specific and significant expression in adult antennae of both sexes [24].

The greater wax moth, *Galleria mellonella* Linnaeus (Lepidoptera: Pyralidae), is an important pest of honeybee products [25]. Larvae of *G. mellonella* use honeycombs to make silken galleries, disrupting the development and growth of bees; this event is called Galleriasis [26]. Interestingly, its mating is highlighted by males producing an acoustic signal and a sex pheromone blend, mainly formed by aldehydes nonanal and undecanal (major components) [27] that attract conspecific females. These aldehyde-type compounds have been reported from different sources, such as insect sex pheromone [17,28,29,30,31], VOCs from plants [32,33,34], as well as in beehive products, such as honey, pollen, wax, and propolis [35,36,37,38]. The usual methods to manage this pest are based on chemical insecticides, e.g., naphthalene, methyl bromide, paradichlorobenzene, and carbon dioxide (CO_2_); however, these compounds (except CO_2_) represent a health risk and lead to residues in honeybee products [25,39]. Furthermore, no environmentally friendly control methods have been reported for *G. mellonella* thus far. Whereas research in OBPs, ORs, and other olfactory proteins, such as ionotropic receptors (IRs), gustatory receptors (GRs), and chemosensory proteins (CSPs) has been conducted in *G. mellonella* from transcriptome [40,41], no ODEs have been studied in depth. Therefore, the understanding of semiochemical degradation mechanisms by AOXs in antennae of *G. mellonella* would provide the necessary information to corroborate or reject the use of these enzymes as targets through their inhibition and, subsequently, disruption of chemical communication for insect pest control. In this work, we take advantage of transcriptomic data from *G. mellonella* obtained in our laboratory where some aldehyde oxidase enzymes might be involved in the degradation of pheromone components, as well as beehive-derived volatiles. Here, we report the phylogenetic relation, relative expression, and enzymatic activity of two novel AOX-encoding genes, *GmelAOX2* and *GmelAOX3*, from the antennae of *G. mellonella*.

## 2. Materials and Methods

### 2.1. Insect Rearing 

Wild *G. mellonella* were obtained from honeycombs located in the Quepe sector of La Araucanía region, and their rearing was established according to the methodology reported by Zamorano [42] using a diet based on a mixture of sugar in freshly boiled distilled water, to which glycerin and vitamins were added. The food source was based on cereals Nestum^®^ and wheat germ, both mixed in the proportions suggested by the same author. Foster boxes (plastic) were arranged with a rectangular top window, covered by a mesh bonded with silicone, and stored in a growth chamber (ShelLab) at a temperature of 28 ± 1 °C. Once eggs were obtained from moths, they were disposed in control tape towards the diet, facilitating feeding and larval development. Larvae were individualized in plastic pots until reaching the adult stage.

### 2.2. Collection of Honeycomb Volatiles

Volatiles were trapped by using four sterilized borosilicate glass chambers [43]. Honeycomb pieces of 5 cm × 5 cm were introduced and placed at the bottom of the chambers. In two upper outlets (per chamber) were Porapak-Q (Divinylbenzene/Ethyl vinyl benzene) columns (100 mg). A positive/negative pressure air system was used according to Agelopoulos et al. [44]; the air was dried and purified before passing through the glass chamber. The trapping of volatile compounds was carried out for 24 h. Then, compounds were desorbed from the Porapak-Q column with 1 mL of hexane and concentrated up to 50 µL under nitrogen (N_2_) flow [45].

### 2.3. Honeycomb Volatile Identification by GC/MS

The volatile compounds (1 µL) were analyzed using a gas chromatograph (Thermo Scientific Trace 1300, Milan, Italy) coupled to a mass spectrometer (GC/MS) (Thermo Scientific ISQ 7000) equipped with an HP-5 (5% cross-linked phenyl-methyl siloxane) capillary column (30 m, 0.25 mm, 0.25 µm). Helium (He) was used as the carrier gas, with a flow rate of 1 mL/min. Mass spectrum acquisition was performed in the mass range from 30 to 500 m/z. Ionization was performed by electron impact at 70 eV with an ion source at 250 °C. The GC oven was programmed to remain at 40 °C for 2 min and increase by 4 °C/min to 250 °C, holding for 5 min. The temperatures of the GC injector, transfer line, and detector were 250 °C [45]. Tentative structural assignments were made by comparing their mass spectra with the MS library (NIST), as well as by comparison of their Kovats indices by using the *n*-alkanes (C_9_-C_21_) and (C_21_-C_40_) series with Kovats indices published from the literature and the injection of standards (Sigma-Aldrich, St. Louis, MO, USA).

### 2.4. Identification of AOX Transcripts by Comparing Two Transcriptome Data of G. mellonella

The AOX identification was assisted by using the whole-head (head plus antennae) transcriptome for *G. mellonella* [41] assembled in our laboratory, and the antennal transcriptome assembled by Zhao et al. [40]. Firstly, an in-house database of lepidopteran AOXs created with sequences reported in the literature was used in order to make a local BLAST through the makeblastdb script for nucleotide and protein with the data set obtained from the assembled transcriptomes. Subsequently, transcripts were identified by local searches using the Tools of NCBI BLASTx and BLASTn [46] between our database and the assembled data set of *G. mellonella*. BLAST hits with e-values < 1.0 × 10^−5^ were considered to be significant [47], and genes were assigned to each contig based on the BLAST hits with the highest score value. The open reading frame (ORF) of each unigene was determined by using the ORF finder tool (https://www.ncbi.nlm.nih.gov/orffinder, accessed on 25 July 2022), and sequences with >1000 amino acids (aa) were selected. These sequences were used as a database in order to compare them with the transcriptome assembled by Zhao et al. [40], and the identification was carried out as mentioned above. In addition, the InterPro platform was used to evaluate their gene ontology (GO), and sequences were submitted to Expasy to calculate their molecular weight in silico.

### 2.5. Phylogenetic Analysis of G. mellonella AOXs

The phylogenetic analysis was carried out using identified AOX transcripts in *G. mellonella*, and sequences from a study on AOXs Lepidoptera [48]. Full-length aa sequences that include conserved domains were aligned using MAFFT server7 [49]. GUIDANCE2 server8 was used to check the consistency of the multiple sequence alignment [50]. Briefly, the consistency of the alignment was measured with a score less than 0.5, in which sequences were deleted. It is worth noting that confidence scores near 1 and 0 suggest a highly and poorly consistent alignment, respectively. Finally, phylogenetic analysis was performed using the maximum-likelihood method with FastTree software [51]. To highlight clades, specific taxa, and functional evidence, the phylogenetic tree was edited using FigTree software9 and image editor Inkscape 0.48 software.

### 2.6. Total RNA Extraction, cDNA Synthesis, and Primer Design

Total RNA extraction was performed following the methodology proposed by Gu et al. [52], using different tissue samples (antennae, n = 100; legs, n = 50; wings, n = 50; bodies without legs, wings, and heads, n = 10) from males and females extracted with TRIzol reagent (Invitrogen, Carlsbad, CA, USA). Moreover, the RNA concentration was analyzed using a Quantus Fluorometer (Promega). The RNA integrity was checked by 1% agarose gel electrophoresis, and samples were stored at −80 °C until use. From the RNA samples, through the semi-quantitative RT-PCR technique and using a thermocycler (GeneTechnologies), a stock of cDNA was generated at a concentration of 100 ng/µL for each tissue. AffinityScript qPCR cDNA Synthesis Kit (Stratagene, Cedar Creek, TX, USA) was used following the manufacturer's instructions. Finally, primers used for every AOX transcript were designed using the PrimerQuest^®^ program (IDT, Coralville, IA, USA).

### 2.7. Analysis of Tissue Distribution by Semi-Quantitative RT-PCR

Amplification of *GmelAOXs* in several tissues (antennae, bodies, legs, and wings) from males and females was carried out following Gu et al. [52] and Lizana et al. [41], and the PCR mix is mentioned in Appendix A. The housekeeping gene *β*-actin (accession code *KP331524*) was used as an endogenous gene and positive control for the analysis. The PCR program for *β*-actin was performed under the following conditions: an initial denaturation step of 95 °C for 2 min, followed by 35 cycles of (1) denaturation step of 95 °C for 30 s, (2) annealing step of 48 °C for 30 s, (3) extension of 72 °C for 1 min, and a final extension for 10 min at 72 °C. PCR products were analyzed on 1% agarose gel and visualized after staining with SYBR™. The PCR program for AOXs consisted of an initial denaturation step of 95 °C for 3 min, followed by 35 cycles of (1) denaturation step of 95 °C for 30 s, (2) annealing step of 50 °C for 30 s, (3) extension of 72 °C for 1 min, and a final extension for 10 min at 72 °C.

### 2.8. Analysis of GmelAOXs Relative Expression by qRT-PCR

All qRT-PCR reactions were performed using Brilliant II SYBR Green qPCR Master mix in qPCR-compatible equipment. The following cycling conditions were used: 95 °C for 10 min, followed by 40 cycles at 95 °C for 30 s, 57 °C for 30 s, and 72 °C for 1 min. The presence of a specific amplified PCR product was verified for each reaction by melt curve analysis, with 95 °C for 15 s, 55 °C for 1 min, and 95 °C for 15 s. The specific primers used in this study were designed using the PrimerQuest^®^ program (IDT, Coralville, IA, USA), and their efficiencies (ranging from 90% to 110%) were validated by standard curve with five 10× serial dilutions of antennal cDNA. The housekeeping gene *β*-actin was used as an internal control. All the experiments were performed using three biological replicates, each with three technical replicates. The relative quantification was analyzed by the 2^−ΔΔCt^ based on the Pffafl method [53]. Statistical differences between tissues from males and females were analyzed by one-way ANOVA with Tukey’s test (*p* < 0.05) using SPSS Statistics 22. Semi-quantitative PCR and qRT-PCR primers are listed in Appendix A.

### 2.9. AOX Activity from Antennal Extract

Enzymatic activity was performed using different aldehyde substrates according to Wang et al. [54]. Thiazolyl blue tetrazolium bromide (MTT) was used as the electron acceptor and phenazine methosulfate (PMS) was the electron donor. Then, the enzymatic activity was measured according to the purple insoluble MTT formazan formation. Protein concentration was measured by the Bradford method using bovine serum albumin as a quantitative standard. The reaction contained fresh crude antennal extracts (75 µg), 3 mM aldehyde substrate dissolved in DMSO, 0.1 M potassium phosphate buffer (pH 8.0), 0.4 mM MTT, and 0.1 mM PMS. Then, it was incubated at 30 °C for 1 h, and the reaction was quenched with 10% acetic acid. The reaction was determined at 570 nm using a UV-5100B spectrophotometer (Metash Instruments, Shanghai, China). All experiments were performed in triplicate.

## 3. Results

### 3.1. Volatile Organic Compounds in Honeycombs

Appendix A contains the complete profile of volatiles according to retention time and their structure class. A total of 74 VOCs were identified from infested honeycombs. For instance, terpenes such as *α*-pinene, camphene, *β*-pinene, *α*-terpinene, limonene, *γ*-terpinene, linalool, 3-terpineol, limonene-1,2-diol, solongifolene, (*E*)-thujopsene, *α*-caryophyllene, *β*-ionone, (+)-*β*-selinene, *α*-muurolene, *β*-bisabolene, *α*-cadinol, (*Z*,*Z*)-farnesol, murgantiol, and sclarene were identified. In addition, several esters were found, such as hexyl acetate, ethyl phenylacetate, n-octyl acetate, linalyl butyrate, isobornyl butyrate, 3-octyl tiglate, methyl cycloundecanecarboxylate, isobutyl decanoate, ethyl tetradecanoate, 3-hexenyl-(*Z*)-cinnamate, 10-undecenyl angelate, ethyl-(*Z*)-7-hexadecenoate, (*Z*)-4-hexadecenyl acetate, incensole acetate, methyl-(*Z*)-communate, 2-ethylhexyl-(*E*)-4-methoxycinnamate, and integerrimine. Other types of compounds were identified, such as alcohols, ketones, ethers, alkanes, furans, carboxylic acids, and phenols. Likewise, aldehyde-type compounds were found, including octanal, nonanal, (*E*)-2-nonenal, undecanal, (*Z*)-2-dodecenal, *α*-sinensal, (*Z*)-10-hexadecenal, and (*E*,*E*,*Z*,*Z*)-4,6,11,13-hexadecatetraenal (Table 1).

### 3.2. AOX-Related Transcripts Obtained by Comparing Two Transcriptomes

According to the results from the comparison of the whole-head transcriptome assembled in our laboratory and the antennal transcriptome reported by Zhao et al. [40], it was possible to identify two AOX transcripts. From BLASTp analysis, sequences with unigenes *DN3568* and *DN34847* were matched with two aldehyde oxidases (accession code *QPF77599.1* and *QPF77600.1*) of *G. mellonella*, which were retrieved from the NCBI database with identity percentages of 99.13% and 99.18%, respectively. These sequences were annotated as *GmelAOX2* and *GmelAOX3*, and can be found in Appendix A. Thus, *GmelAOX2* and *GmelAOX3* had ORFs of 3816 and 3675 nucleotides (nt), respectively. In addition, both aa sequences presented the three typical domains of AOXs, namely (1) two 2Fe-2S clusters, (2) FAD-binding, and (3) molybdenum cofactor (Moco) binding, based on a search of the InterPro platform. On the other hand, the results of the theoretical molecular weight in these enzymes according to the Expasy platform showed that *GmelAOX2* and *GmelAOX3* have subunits of 141 and 134 kDa, respectively. According to phylogenetic analysis (Figure 1), only *GmelAOX2* has a linage related to the clade with ODE function, and is closely associated to *AtraAOX2* from the navel orangeworm *Amyelois transitella* Walker, with a high bootstrap value (>70%).

### 3.3. Tissue Distribution and Relative Expression Levels of GmelAOX2 and GmelAOX3

Our results suggest that enzymes do not have a tissue-specific expression; *GmelAOX2* and *GmelAOX3* are slightly expressed in the male body, and the latter is also expressed in the female body (Figure 2). Besides body, PCR products of both enzymes were presented in male rather than female antennae. On the other hand, there were significant differences in relative expression of *GmelAOXs* between tissues according to qRT-PCR results (Figure 3). We found two *GmelAOX* genes (*GmelAOX2* and *GmelAOX3*) enriched in antennae compared to other tissues tested, and the level of these genes in male antennae was significantly higher than in female antennae.

### 3.4. AOXs Activity from Antennal Extract

In order to evaluate the enzymatic activity of GmelAOXs in antennal extracts, some aldehydes of those previously identified from honeycombs, as well as pheromone components, were selected. It is worth noting that some aldehydes were not commercially available, and other structurally similar compounds were included, such as *trans*-2-hexenal, hexenal, and decanal. Moreover, undecane was included as the corresponding alkane of undecanal. The activity was presented as relative activity (%), where benzaldehyde was used as a standard (100%) according to Choo et al. [29]. Figure 4 shows the results obtained, where undecanal had the highest activity, followed by decanal, undecane, nonanal, hexanal, *trans*-2-hexanal, and octanal. Notably, undecane showed the third-highest activity even without presenting an aldehyde in its structure.

## 4. Discussion

Here, we identified several compounds in honeycombs infested with *G. mellonella*. Among these chemicals, nonanal and undecanal are reported as part of its sex pheromone blend [25,63]. Likewise, other aldehydes are associated with bees and their products, such as octanal from honey and wax [55,56,57], (*E*)-2-nonenal with propolis [60], and *α*-sinensal with honey [35]. On the other hand, (*Z*)-2-dodecenal is a volatile found in clementine oil [64], (*Z*)-10-hexadecenal has been reported as a pheromone in some lepidopteran of the Crambidae family [65,66], and (*E*,*E*,*Z*,*Z*)-4,6,11,13-hexadecatetraenal has been identified as the major sex pheromone of *C. promethea* [17].

*G. mellonella* is a ubiquitous pest to honeycombs and a nocturnal insect which lays eggs inside honeycombs, especially when bees are less active [67]. Moreover, when adults emerge from the pupa stage, they fly toward the trees to mate. So, this moth must rely on its olfactory system for detecting and decoding semiochemicals, and thus define its behavior. The clearance of compounds that remain in the sensillar lymph is a critical step in the olfaction process, which is carried out by ODEs, in order to facilitate the entrance of new stimuli in the antennae [4]. Research around ODEs is limited, where most studies have been reported in Lepidoptera, namely the antennal esterase *SlCX7* in the cotton leafworm *Spodoptera littoralis* Boisduval, which can hydrolyze the pheromone and plant compounds [68]. In addition, Ref. [69] identified an ODE gene from the polyphagous moth *S. exigua* Hübner, namely *SexiCXE10*, which showed high activity for ester plant volatiles. In terms of AOXs, studies are even scarcer. Bioinformatic analyses have served to identify putative enzymes; for instance, four AOXs were reported in *C. medinalis*, three mainly found in the adult abdomen and one enriched in antennae [23]. Another author identified three AOXs in the pink borer *Sesamia inferens* Walker; two were antennae-specific (*SinfAOX1* and *SinfAOX2*) and one (*SinfAOX3*) was expressed in antennae and the abdomen [30]. The phylogenetic relationships of these enzymes have allowed the clustering of AOXs with ODE function [48]. Phylogenetic analysis showed that both *GmelAOX2* and *GmelAOX3* have evolved from xanthine dehydrogenases (XDHs) (Figure 1), similarly to other reported AOXs [23,70], due to gene duplication events [71]. Remarkably, *GmelAOX2* was grouped in the clade with ODE function, where other functionally studied AOXs [29,54,72] are found, shedding light on its role as an ODE.

As a result of semi-quantitative RT-PCR, tissue-specific expression was not performed because *GmelAOX2* was expressed in the body and antennae of males, and *GmelAOX3* was expressed in the body of both sexes and male antennae only. However, we suggest that at least *GmelAOX2* has a sex-biased expression in male antennae according to RT-PCR, despite its slight expression in the body. The expression of these enzymes in the body might be associated with a detoxification process, due to the degradation of xenobiotic compounds such as pesticides [70]. In fact, AOXs are able to use several compounds as substrates, i.e., N-heterocyclics, N-oxides, azo dyes, and aldehydes with different hydrocarbon chain sizes [73]. In contrast to the expression of AOXs in the body, their expression in antennae would be involved in the degradation of sex pheromone components or aldehyde-type plant volatiles, being potentially classified as PDEs [24]. Huang et al. [70] showed an expression of two AOXs in *C. pomonella* (*CpomAXO1* and *CpomAOX2*) in several tissues (antennae, thoraxes, abdomens, legs, and wings), but they propose a role in odorant degradation when these enzymes are expressed in antennae even if they are not specific to this tissue. Insect antennae are the main olfactory organs; therefore, the expression of AOXs in these structures would be closely related to the degradation process of semiochemicals [24]. Thus, studies in *S. inferens* and *C. medinalis* have shown significant differences in the relative expression of these AOXs (*SinfAOX1*, *SinfAOX2*, and *CmedAOX2*) according to qRT-PCR, where male antennae showed a higher expression [23,30]. Here, *GmelAOX2* and *GmelAOX3* showed a higher expression in males (Figure 3). Interestingly, it is common for conspecific females to synthesize and release the sex pheromone to attract males [74]; however, in *G. mellonella*, males produce the sex pheromone [25]. Therefore, a higher expression of AOXs in male antennae could help to rapidly degrade aldehyde-type compounds that are part of the sex pheromone blend of *G. mellonella*, allowing the entrance of other chemical cues. Notably, low mRNA expression does not always imply low protein levels or low enzyme activities [75]. Preliminarily, these results indicate that both aldehyde oxidases in *G. mellonella* could be classified as general odorant-degrading enzymes (GODEs) instead of PDEs.

The AOX function in insects towards semiochemical degradation has been recently studied from pheromone gland extracts of *B. mori* (*BmorAOX5*), where they were active over several aldehydes (i.e., benzaldehyde, salicylaldehyde, vanillic aldehyde, heptanal, and propanal) [72]. Furthermore, Wang et al. [54] evaluated the activity of a recombinant AOX (*PxylAOX3*) from the diamondback moth *Plutella xylostella* Linnaeus, where it was capable of degrading its sex pheromone components and several aldehyde-type volatiles derived from plants. Antennal extracts were capable of metabolizing aldehyde-type compounds, such as some of those identified from infested honeycombs. Data published by Leyrer and Monroe [27] indicate that *G. mellonella’s* sex pheromone blend has two major components, nonanal and undecanal, in a ratio of 7:3. Taking this into account, the strongest activity on undecanal (4-fold) compared to benzaldehyde (control) in antennal extracts is consistent. Although nonanal is present in a greater proportion than undecanal, 1.7-fold higher enzyme activity compared with control was obtained. Moreover, hexanal, octanal, and decanal, reported as minor sex pheromone components of *G. mellonella* [25], performed more activity than control. Studies show that aldehyde-type VOCs have insecticide activity against some dipterans, such as hexanal and octanal as toxic compounds to *Drosophila sechellia* Tsacas and Baechli and *D. melanogaster* Meigen, respectively [76]. In addition, decanal has shown toxicity against the scarab beetle *Tribolium castaneum* Herbst [77], and nonanal has shown toxicity against *S. frugiperda* Wlaker [78]. Undecanal was also reported as a repellent against mosquito *Anopheles gambiae* (s.l.) [79]. The degradation of these compounds by *GmelAOXs* shed light on the key role in the olfaction process of this insect. The fact that males present a higher expression of AOXs supports a metabolization of their own pheromone components. The antennal extracts also showed activity over *trans*-2-hexenal, a compound also reported as an insecticide to *T. castaneum* [80]. Although undecane was surprisingly active in antennal extracts, it is likely that other enzymes are present in the antennae, i.e., monooxygenases [81,82], that are capable of transforming the alkane to a corresponding alcohol (undecanol), and then it is dehydrogenated to the corresponding aldehyde (undecanal) by alcohol dehydrogenases [30,70]. Bearing this in mind, this study shows that AOXs presented in antennal extracts were able to metabolize several compounds. *GmelAOX2* could play a more active role in the degradation of aldehyde-type compounds compared to *GmelAOX3* due to its phylogenetic relationship with *AtraAOX2* from the pyralid moth *A. transitella*. Moreover, we have not had success in the expression of the active forms of both enzymes GmelAOX2 and GmelAOX3 by using a bacterial expression system. Other heterologous expression systems (e.g., baculovirus expression system) could be evaluated in further studies in order to corroborate a pheromone-biased degradation of these enzymes.

In conclusion, we have demonstrated that two novel AOX transcripts are expressed in *G. mellonella*, and *GmelAOX2* has a sex-biased expression. These enzymes have a crucial role in metabolizing sex pheromone compounds as well as plant-derived aldehydes, which are related to honeycombs and the life cycle of *G. mellonella*. Thus, new strategies for integrated pest management are required, and in the case of *G. mellonella*, the search for antagonist compounds capable of inhibiting AOXs represents an alternative.

## Figures and Tables

**Figure 1 insects-13-01143-f001:**
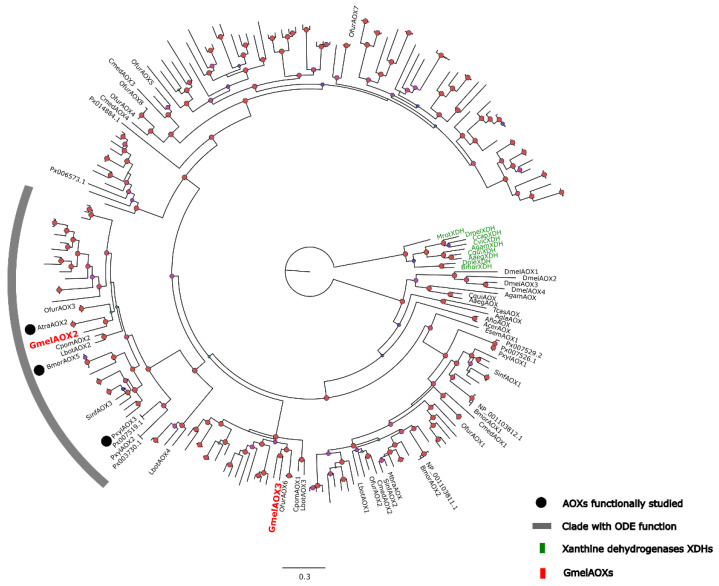
Phylogenetic tree of AOXs identified in *G. mellonella* transcriptomes as well as sequences from a previous report in lepidopteran [48]. Black circles show functional AOX reported in *A. transitella* (*AtraAOX2*), *P. xylostella* (*PxylAOX3*), and *B. mori* (*BmorAOX5*). Gray line shows AOXs clustered in the clade with ODE function. In green are the xanthine dehydrogenases (*XDHs*) and in red are *GmelAOXs*. Confidence scores are indicated as red circles (>70%) in nodes.

**Figure 2 insects-13-01143-f002:**
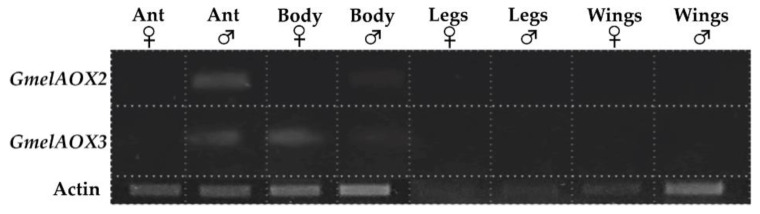
Electrophoresis with 1% agarose gel. *Galleria mellonella* aldehyde oxidase (AOX) transcript levels in different tissues tested by semiquantitative RT-PCR. All reactions were run under the same experimental conditions. Ant., antennae; Body (without head, wings, and legs); Legs; Wings. (

) female; (
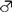
) male.

**Figure 3 insects-13-01143-f003:**
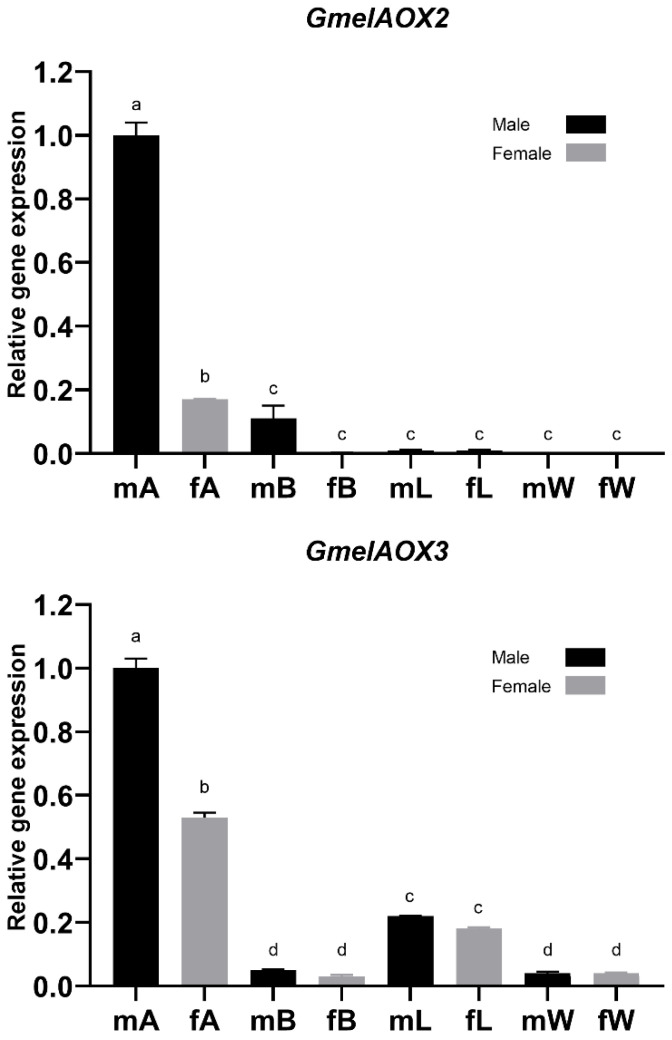
*Galleria mellonella* aldehyde oxidase (AOX) transcript levels in female and male antennae tested by qRT-PCR. Black and gray bars indicate males and females, respectively. Different lowercase letters indicate significant differences (one-way ANOVA with Tukey’s test, *p <* 0.05). Male antennae, mA; female antennae, fA; male body, mB; female body, fB; male legs, mL; female legs, fL; male wings, mW; female wings, fW.

**Figure 4 insects-13-01143-f004:**
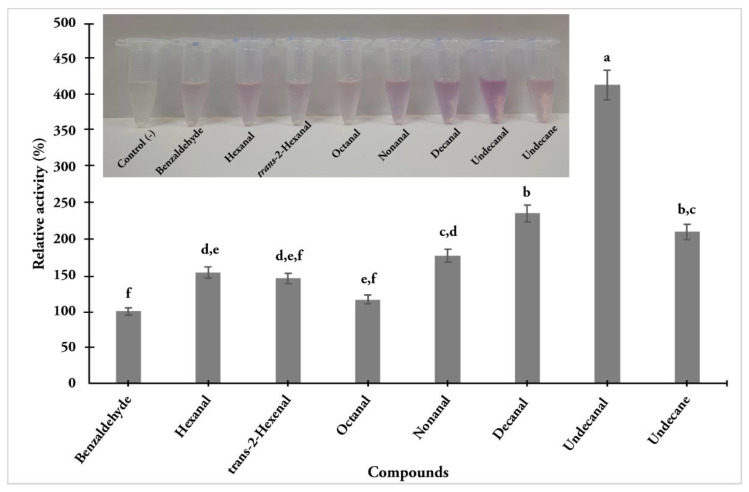
Relative activity of substrates oxidation by antennal extracts of *G. mellonella*. The oxidation of substrates (3 mM) by antennal extract (75 µg) was determined at 30 °C and quenched with 10% acetic acid. The reduction in MTT was measured spectrophotometrically at 570 nm. Activity is observed according to the purple insoluble MTT formazan formation. A blank was set as a negative control by adding buffer only. ANOVA was followed by Tukey’s test for multiple comparisons for an average mean comparison. Different letters indicate significant differences (*p* < 0.05).

**Table 1 insects-13-01143-t001:** Honeycomb volatiles identified by GC/MS.

Compound	Source	Source Reference	ng/Kg	RT (min)	K_exp_	K_ref_	Identification Reference
**Octanal**	Honey, honeybees, wax	[55,56,57]	1.45	12.69	999	999	Standard *; [58]
**Nonanal**	Honeybees, *G. mellonella* pheromone, wax	[25,55,59]	22	15.27	1085	1084	Standard *
**(*E*)-2-Nonenal**	Propolis	[60]	-	17.26	1153	1155	[58]
**Undecanal**	Honeybees, *G. mellonella* pheromone	[25,59]	35.3	21.00	1284	1284	Standard *
**(*Z*)-2-Dodecenal**	-	-	-	25.70	1467	1467	[61]
***α*-Sinensal**	Honey	[35]	-	32.16	1752	1752	[62]
**(*Z*)-10-Hexadecenal**	-	-	-	33.27	1804	1804	[61]
**(*E,E,Z,Z*)-4,6,11,13-Hexadecatetraenal**	-	-	-	35.78	1926	1926	[17]

RT= Retention time obtained from the mass spectrums; K_exp_ = Kovats determined by using the n-alkane series (C_9_-C_21_ and C_21_-C_40_); K_ref_ = Kovats based on the injection of Standards (*) and search in literature by the comparison with GC/MS spectrometry library. (-) = Non reference on source of the tentative compounds. Semiquantification (ng/Kg) was made by the comparison with area peaks of standards (50 ppm).

## Data Availability

The data presented in this study are available in article or Appendix A.

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
