# Peer review of "Characterization of Two Aldehyde Oxidases from the Greater Wax Moth, Galleria mellonella Linnaeus. (Lepidoptera: Pyralidae) with Potential Role as Odorant-Degrading Enzymes"

_insects, 2022, doi:10.3390/insects13121143_

Round 1

Reviewer 1 Report (Previous Reviewer 1)

I think this manuscript was improved a lot compared to the last version. But, I still have several questions:

1.      Although the author has added the PCR gel including the reference genes in RT-PCR result. But the intensity of actin bands of legs of female and male, and wings of females was significantly less than other tissues, which means that the reference gene was not perfect or the concentration of cDNA templates was different.

2.      The author still didn’t answer the question that the authors just investigated the enzymatic activity of antennal extracts to metabolize the sex pheromone compounds and plant-derived volatile in the manuscript, but these results could not provide any evidence for the roles of GmelAOX2 and GmelAOX3 in the odorant degradation as many different protein species exist in the antennal extracts. It will be better to perform heterologous expression of these two genes and investigate the enzymatic activity of these two purified proteins on odorant degradation, which can make the results more convincing.

Author Response

Please see the attachment,

Kind regards.

Reviewer 2 Report (New Reviewer)

These are my main comments on the manuscript (insects-1945984) entitled “Characterization of two aldehyde oxidases from the greater wax moth, Galleria mellonella (Lepidoptera: Pyralidae) with potential role as odorant degrading enzymes”. The manuscript investigates two novel aldehyde oxidases, AOXs-encoding genes (GmelAOX2 and GmelAOX3) using transcriptomic analysis in Galleria mellonella. GmelAOX2. Following moderate revisions should be incorporated in the manuscript prior to acceptance.

1. I have concerns about the manuscript sections that I believe need to be addressed in order to improve its clarity.

2. A hypothesis for this work is needed.

3. Other revisions could be checked in PDF attached.

Author Response

Response to Reviewer 2 Comments

Point 1: I have concerns about the manuscript sections that I believe need to be addressed in order to improve its clarity.

Response 1: We thank the reviewer suggestion. We have modified some sections in the manuscript, and we have made the changes suggested in the pdf. attached. On the other hand, sensillar lymph is used in several insects studies (Vogt & Riddiford 1981; Leal, 2013; Zhang et al., 2014).

Point 2: A hypothesis for this work is needed.

Response 2: We thank the reviewer comment. We have added a hypothsis in the manuscript (see line 122).

Please see the attachment (draft)

Round 2

Reviewer 1 Report (Previous Reviewer 1)

The authors answered all my concerned questions, it can be accepted in the present form. But i still strongly suggested the authors to perform gene expression to identify the enzyme functions and provide more evidence for the conclusions in the future study.

This manuscript is a resubmission of an earlier submission. The following is a list of the peer review reports and author responses from that submission.

Round 1

Reviewer 1 Report

In this study, the authors identified two odorant degrading enzymes, GmelAOX2 and GmelAOX3 in the greater wax moth, Galleria mellonella. The transcript levels of GmelAOX2 and GmelAOX3 were investigated in different tissues and antennae females and males using RT-PCR and RT-qPCR. Functional assay revealed that antennal extracts had the strong enzymatic activity against undecanal which was characterized in the VOCs trapped from honeycombs. Although the study is interesting for readers, there are problems with the experiment.

There are several comments:

1.   The Figure 1 is unclear, please provide image with high resolution.

2.  RT-PCR. It will be better to add the PCR gel of reference genes to explain that the concentration of cDNA template is the same for all treatments. Also, I would suggest to add the RT-qPCR results for the expression levels of genes in different tissues.

3.   The authors examined the enzymatic activity of antennal extracts to metabolize the sex pheromone compounds as well as plant-derived volatile, but these results could not direcltly demonstrate the roles of GmelAOX2 and GmelAOX3 in the odorant degradation. I would suggest the author to perform heterologous expression of these two genes and investigate the enzymatic activity of these two purified proteins on odorant degradation, which can make the results more convincing.

Reviewer 2 Report

The study presented by Godoy et al is very interesting and presented in a well-mannered form. Though in my view the manuscript should be accepted in the present form, the authors need to improve the quality of Figures 2 and 4. The bands are not clearly visible. I strongly recommend the acceptance of the manuscript.